# Fruit and Vegetable Consumption in Mothers and Their Children Aged 4–10 and Its Relationship with Maternal Feeding Practices

**DOI:** 10.3390/nu17060941

**Published:** 2025-03-07

**Authors:** Marzena Jeżewska-Zychowicz, Aleksandra Małachowska, Marta Sajdakowska

**Affiliations:** 1Institute of Human Nutrition Sciences, Warsaw University of Life Sciences (SGGW-WULS), Nowoursynowska 159 C, 02-776 Warsaw, Poland; marzena_jezewska_zychowicz@sggw.edu.pl; 2Military Institute of Medicine—National Research Institute, Zegrzyńska 8, 05-119 Legionowo, Poland; ola.malach@gmail.com

**Keywords:** family, child behavior, fruit, vegetables, feeding-related practices

## Abstract

The low intake of fruit and vegetables among the Polish population prompts a search for factors that can be addressed in dietary interventions, such as the family food environment. **Background/Objectives**: The objective of this study is to explore the link between maternal feeding practices applied to children aged 4–10 years and fruit and vegetable intake among children and their mothers. **Methods:** A cross-sectional study using a Computer-Assisted Web Interview technique took place in 2020–2021 among 260 Polish women who were mothers of children aged 4–10. Statistical analysis included descriptive statistics, Spearman’s correlation coefficient, and multiple linear and logistic regression. **Results:** The mother’s intake of fruit was strongly correlated with the children’s intake of fruit (β = 0.309; *p* < 0.001), and a similar correlation was found for vegetable intake (β = 0.428; *p* < 0.001). Apart from the mother’s fruit and vegetable intake, the Food as a Reward practice correlated negatively (β = −0.164; *p* = 0.015), while Monitoring (β = 0.158; *p* = 0.017) and Modeling (β = 0.170; *p* = 0.028) correlated positively with vegetable intake in children. The Monitoring practice correlated positively (β = 0.221; *p* < 0.001) with children’s fruit intake. After adjusting for the mother’s age, child’s gender, and mother’s recommended intake of fruit and vegetables, the Monitoring practice (OR = 1.971; *p* = 0.025) positively correlated with meeting the daily recommendations of fruit and vegetables in children. However, the Food as a Reward (OR = 0.484; *p* = 0.018) and Emotion Regulation (OR = 0.345; *p* = 0.008) practices negatively correlated with meeting the daily recommendations of fruit and vegetables in children aged 4–6, while the Monitoring practice (OR = 4.141; *p* = 0.017) correlated positively with meeting the daily recommendations of fruit and vegetables in children aged 7–10. **Conclusions:** The findings have shown that the mother’s fruit intake strongly correlates with the child’s fruit intake. Moreover, some maternal feeding practices, i.e., the Food as a Reward and Emotion Regulation practices, were negatively associated with meeting fruit and vegetable recommendations in younger children, while the Monitoring practice was positively related to meeting them in older children.

## 1. Introduction

Eating behaviors have a strong genetic basis [1]. Nevertheless, the environment in which children live and grow up also plays a key role in conditioning their behavior [2]. Among the environmental determinants of eating behaviors, the early feeding environment and parental feeding practices are the most extensively studied [3]. Throughout childhood, parents are responsible for shaping their children’s food environment through the availability of foods and the rules they establish around the timing, frequency, and structure of meals [4]. As significant providers, models, and regulators of food intake, parents significantly influence their children’s eating habits in various ways [5]. When feeding the child, parents use multiple strategies to control or modify what, when, and how much their child eats, in this way predisposing children to favorable or unfavorable behaviors [6,7]. Parental feeding practices (PFPs) include pressuring a child to eat more, restricting certain foods, or monitoring their food consumption [4]. Parental influence on children’s behaviors is complex due to the dynamic relationship between parents and their children. Parents often adapt feeding practices based on their children’s characteristics (i.e., temperament, weight, or eating behaviors) [7].

Parents, as models, influence the child’s behavior through what and how much they eat. Therefore, parents’ consumption of vegetables and fruits can be crucial in conditioning the child’s consumption of these foods [8,9]. The results of previous studies confirm the differences between mothers and fathers regarding parenting feeding practices and their impact on children’s eating behaviors, i.e., fathers use more pressure to eat, fewer rewards, and more restrictions, but also impose fewer restrictions on snacks [8,9,10,11]. Thus, this underlines the importance of studying the individual role of each parent in child-feeding research [8]. In Poland, women are still more involved in childcare than men, which may significantly influence children’s eating habits through their own behavior and the feeding practices applied to the child [11].

Although it is improving, vegetable and fruit intake is still too low across all age groups in Poland, similarly to most American, Australian, and European countries [12,13]. Research has found that fruit intake decreases from age 7 and reaches its lowest level during adolescence. On the other hand, vegetable intake changes little during childhood and adolescence [14]. Vegetable and fruit consumption among children correlates with parent’s education level [15] and their body mass index [16]. Other factors include the age of the child [17], socioeconomic status [18,19], fruit and vegetable consumption of parents, food availability at home [10,20,21], and the frequency of family meals [22]. Research indicates that parents’ eating behaviors and parental feeding practices might affect children’s eating habits and encourage healthy eating behaviors, such as a higher intake of fruit and vegetables [23,24,25] and a lower intake of low-nutrient-dense foods [26], leading to greater diet quality. However, there are still some inconsistencies in the results concerning various parental feeding practices and fruit and vegetable intake, but they are potentially modifiable factors that can be targeted through parenting interventions [3].

A clear understanding of how paternal feeding practices are associated with children’s eating behaviors is essential for educators, parents, and clinicians who aim to promote healthy eating habits in children. However, there is still a gap in the information regarding the relationship between maternal feeding practices, mothers’ consumption of fruit and vegetables, and Polish children’s eating fruit and vegetable consumption. Since children’s school activity, the long time spent outside the home, and the meals consumed [27] may influence the relationship between their fruit and vegetable consumption and maternal feeding practices, being in kindergarten (age 4–6) or early school (age 7–10) were considered criteria for inclusion in the study. Thus, the objective was to determine the relationship between the consumption of fruit and vegetables among children and their mothers and the maternal feeding practices (MFPs) that mothers used towards their children aged 4–10 years.

## 2. Materials and Methods

### 2.1. Study Design and Sample Collection

The data were collected in Poland in 2020–2021 using a Computer-Assisted Web Interview technique for a cross-sectional study. The invitation to participate in the study, along with a short description of its purpose and a link to the survey, was made available in Warsaw (5 schools and 5 kindergartens) and Ożarów Mazowiecki (1 school and 1 kindergarten). The inclusion criteria were women over the age of 18 living in Masovia province with at least one child aged 4–10 and consenting to participation in the study. The exclusion criteria were being below the age of 18, not living in Masovia province, not having or not exercising legal care over a child aged 4–10, and not consenting to participation in the study. The mother was asked to indicate the child (if there were two or more children in the family in this age range) whose fruit and vegetable intake and feeding practices were measured. The age of any other children in the family was not taken into account. The survey lasted no longer than 15 min. Data anonymity and confidentiality were assured. The study sample included 260 respondents who correctly filled out the questionnaire.

The study design received approval from the Ethics Committee at the Institute of Human Nutrition Sciences, Warsaw University of Life Sciences, Poland (Resolution No. 29/2020; approval date: 24 August 2020). This study was conducted following the guidelines of the Declaration of Helsinki.

### 2.2. Questionnaire

Maternal feeding practices (MFPs) were assessed using a Polish adaptation of the Comprehensive Feeding Practices Questionnaire (CFPQ) [28]. The CFPQ allows for a mother-reported measure of the feeding practices applied to the child. Statements from the 7 subscales, i.e., Involvement, Modeling, Monitoring, Emotion Regulation, Food as a Reward, Restriction for Weight Control, and Child Control, were included in the questionnaire. Respondents were asked to report on the different family situations and feeding practices used when feeding the child, with responses ranging from never or disagree (1) to very often or agree (5) [28]. Scores for the subscales were calculated by adding up the individual scores for questions or statements from the questionnaire and counting the mean values. The resulting scores ranged from 1 to 5. The higher the score, the more often a given practice was used by the child’s mother. Cronbach’s alpha reliability coefficients were calculated for each of the subscales. The degree of fit of the first six scales was satisfactory (Cronbach’s alpha: 0.719, 0.774, 0.882, 0.890, 0.766, and 0.879, respectively), while the Cronbach’s alpha for Child Control was unsatisfactory (0.621), thus the subscale was not included in further analysis.

Intake of vegetables and fruit (fresh and processed separately) was assessed using questions derived from the Dietary Habits and Nutrition Beliefs Questionnaire (KomPAN^®^) [29]. Respondents self-reported their food consumption using a 6-point frequency scale from less than once a month or never (1) to 1–3 times a month (2), once a week (3), a few times a week (4), once a day (5), or a few times a day (6). In the same way, they reported on their children’s fruit and vegetable consumption. Based on the frequency scale, the daily frequency was calculated. The values range from 0—less than once a month or never (1)—to 0.06—1–3 times a month (2), 0.14—once a week (3), 0.5—a few times a week (4), 1—once a day (5), or 2—a few times a day (6) [30]. Participants were also asked how many portions of products from each food group they and their children ate daily, given that 1 portion of vegetables and fruit (fresh and processed) equals 100 g. Examples of the portion sizes were added. Food intake was calculated separately for children and their mothers for vegetables and fruit by multiplying the daily frequency of consumption and amount of portions consumed. The adequacy of fruit and vegetable intake was assessed based on the guidelines for the Polish population, which recommend at least 4 portions (400 g) of fruit and vegetables per day [31].

### 2.3. Statistical Analysis

Descriptive statistics, frequency analysis, and cross-tabulations were performed. The Kolmogorov–Smirnov test was used to test the normality of distribution. A chi-square test was used, with an accepted significance level of *p* < 0.05. Spearman’s rank correlation coefficient was used to assess the bivariate correlations between variables. The Wilcoxon signed-rank test was used to confirm differences in the consumption of fruit and vegetables in pairs of mothers and their children.

A multiple linear regression model was applied, adopting the consumption of fruit and vegetables as the dependent variables separately. The independent variables used were maternal feeding practices (Model 1) and maternal feeding practices and mother’s intake (Model 2), and the covariates were the mother’s age and the child’s gender. The following test assumptions were assessed in the linear regression analysis: the normality of the residual distribution (Shapiro–Wilk, *p* > 0.05), lack of multicollinearity (variance inflation factors—VIF < 10), and correlation of residuals (Durbin–Watson test, between 1.5 and 2.5).

Logistic regression analysis was used to search for a correlation between children meeting fruit and vegetable recommendations (dependent variable). Parental practices, i.e., Food as a Reward, Involvement, Modeling, Monitoring, and Restrictions for Weight Control, were independent variables (all continuous variables) included in the model adjusted for the mother and child’s age, child’s gender, and mother’s recommended intake of fruit and vegetables. Odds ratios (ORs) represented the probability of a child belonging to a group with an adequate fruit and vegetable intake (four portions of vegetables and fruit per day or more). The reference groups (OR = 1.00) did not meet the recommendations (less than four portions of vegetables and fruit per day). Wald’s test was used to assess the significance of ORs. The statistical analysis was conducted using IBM SPSS Statistics for Windows, version 29.0 (IBM Corp, Armonk, NY, USA).

## 3. Results

### 3.1. Description of the Study Sample

Table 1 presents the characteristics of the study sample. The sample consisted of 260 respondents who were mothers of at least one child aged 4–10. The median mother’s age was 38 years (IQR = 9 years). The majority of respondents lived in cities with over 100,000 inhabitants. Among the children, 58.1% were girls and 41.9% were boys.

### 3.2. Frequency of Eating Fruit and Vegetables Among Mothers and Their Children Aged 4–10 Years

The frequency of eating fruit and vegetables among the respondents and their children is shown in Table 2. Only 11.2% of mothers and 18.8% of children ate fruit several times a day, while 19.9% of mothers and 19.2% of children ate fruit once a week or less often. Vegetables were eaten several times a day by 6.2% of mothers and 11.9% of children, while 23.8% of mothers and 32.7% of children ate fruit once a week or less often. However, there were no differences in the frequency of fruit and vegetable consumption between the mother and her child (Table 2). There were no differences (*p* = 0.603) in fruit intake between the mother (mean 140.0 g; SD 112.7 g) and her child (mean 136.5 g; SD 111.3 g). The differences in the mother–child pairs were noted in vegetable intake (*p* < 0.001). Mothers ate more vegetables (mean 175.5 g; SD 155.9 g) than their children (mean 108.8 g; SD 115.5 g). Such differences were also noted in total fruit and vegetable intake (*p* < 0.001). Mothers ate more fruit and vegetables (mean 315.5 g; SD 228.0 g) than their children (mean 245.3 g; SD 196.5 g).

### 3.3. Maternal Feeding Practices (MFPs) and Their Relationship with Fruit and Vegetable Consumption in the Study Sample

The descriptive statistics of maternal feeding practices (MFPs) are presented in Table 3.

Monitoring, Involvement, and Modeling (median 4.00) were mothers’ most used feeding practices when feeding their children. The least used maternal feeding practices were Emotion Regulation (2.00) and Food as a Reward (2.33), as well as Restrictions for Weight Control (2.37) (Table 3).

Children’s fruit and vegetable intake correlated positively with Monitoring (r = 0.396 and 0.287, respectively), Modeling (0.286 and 0.278, respectively), and Involvement (0.156 and 0.205, respectively) (Table 4). Their fruit and vegetable intake correlated negatively with Food as a Reward (−0.317 and −0.167, respectively), while vegetable intake also correlated negatively with Emotion Regulation (−0.283). Children’s fruit and vegetable intake frequency correlated positively with Monitoring (0.500 and 0.465, respectively), Modeling (0.396 and 0.465, respectively), and Involvement (0.323 and 0.300, respectively). Their fruit and vegetable intake frequency correlated negatively with Food as a Reward (−0.155 and −0.341, respectively), while vegetable intake frequency also correlated negatively with Emotion Regulation (−0.246). Using Restriction for Weight Control was correlated only with children’s frequency of eating fruit (−0.125) (Table 4).

The results of multiple linear regression showed associations between the maternal feeding practices of Food as a Reward, Modeling, and Monitoring (Model 1) and Food as a Reward, Restrictions for Weight Control, and the mother’s vegetable intake (Model 2) and the consumption of vegetables among children. Table 5 presents the coefficients for the variables in both models. In Model 1, the variable most strongly associated with vegetable intake was the Food as a Reward practice. This MFP (β = −0.240) was negatively related to the intake of vegetables, while Modeling (β = 0.170) and Monitoring (β = 0.158) were positively associated. In Model 2, the mother’s vegetable intake was most strongly positively associated with vegetable consumption in children (β = 0.428). After entering the mother’s vegetable intake into the model, the Food as a Reward practice continued to correlate negatively with children’s vegetable consumption (β = −0.164). There were no more extended associations between the Modeling and Monitoring practices and children’s vegetable intake. However, the Restrictions for Weight Control practice (β = 0.156) showed a positive correlation with children’s consumption of vegetables (Table 5).

When the consumption of fruit was the dependent variable, the results of multiple linear regression showed only one significant association between the Monitoring practice (Model 1) and two associations between the Monitoring practice and the mother’s fruit intake (Model 2) and the children’s fruit intake (Table 6). The Monitoring practice was positively associated with fruit intake in children (β = 0.327 and β = 0.221, respectively). Moreover, in Model 2, the mother’s fruit intake was also positively associated with fruit consumption in children (β = 0.309).

### 3.4. Maternal Feeding Practices (MFPs) and Their Relationship with Meeting Fruit and Vegetable Recommendations in Children

In the study sample, 27.3% of mothers and 49.2% of children had the recommended fruit and vegetable intake. In the group under 6 years, more children (64.4%) than those aged 6–10 (36.0%) had the recommended fruit and vegetable intake. There was also a higher percentage of mothers of younger children (37.0%) who had the recommended fruit and vegetable intake compared to mothers of older children (16.8%) (Table 7).

The results of a logistic regression analysis assessing the effect of maternal feeding practices (MFPs) on children’s likelihood of meeting the fruit and vegetable recommendations in the study sample and two groups identified by the child’s age are presented in Table 8. Only Monitoring practices (*p* < 0.025) were significant in the study sample, but other parental practices were not. The likelihood of the child eating the recommended intake of fruit and vegetables increased by 97.1% when Monitoring practices increased by 1 point.

In children aged 4–6, the likelihood of eating the recommended intake of fruit and vegetables decreased by 51.6% when the Food as a Reward practice increased by 1 point. Moreover, it decreased by 65.5% when Emotion Regulation increased by 1 point. In children aged 7–10, the likelihood of eating the recommended intake of fruit and vegetables increased by approx. 4.1 times when Monitoring practices increased by 1 point (Table 8).

## 4. Discussion

The study results confirmed that the consumption of fruit and vegetables in children aged 4–10 was too low [12,13,32,33]. It appeared that more younger children (under 6 years old) had the recommended fruit and vegetable intake [16,34], which may be caused by the decline in fruit consumption starting from age 7. In contrast, vegetable consumption usually remains more stable [14,34]. Changes in fruit and vegetable intake may be due to a more significant proportion of out-of-home eating, including eating in kindergarten and school. Diversifying the fruit and vegetable supply in these meals and giving children greater freedom to decide how much they eat may favor changes in fruit and vegetable intake [35,36]. Moreover, vegetable consumption in the study group was lower than fruit consumption. This may be because vegetables are the least-liked food [37]. The importance of negative vegetable preference and picky eating [38] in conditioning children’s vegetable and fruit intake shows how crucial early counseling for families with young children is to help appropriately shape their preferences and eating behaviors [39].

Our study showed no differences between mothers and children in fruit and vegetable consumption frequency and fruit intake. In contrast, mothers consumed more vegetables than their children. This may suggest that the mother’s consumption of vegetables may not have been sufficient to encourage her child to eat more vegetables. Nevertheless, the results of multiple linear regression confirmed a positive, although weak, association between mothers’ Modeling practices and children’s vegetable consumption. Such positive effects of using Modeling as a practice to promote vegetable consumption have also been shown in many studies [40,41,42,43]. However, when mothers’ vegetable consumption was introduced into Model 2, the positive association with Modeling practices disappeared, and the strong effect of maternal vegetable consumption was observed. Moreover, there was a lack of a Modeling effect for meeting fruit and vegetable recommendations in younger and older children. Such results may be explained by the fact that vegetable consumption among mothers was relatively low and that, in addition, other factors, e.g., preference and availability [10,20], may have limited the Modeling effect. However, meeting the recommended fruit and vegetable intake was assessed jointly, whereas the Modeling practice did not show any association with fruit consumption among children. This could have contributed to the lack of association between Modeling and meeting the fruit and vegetable recommendations. The lack of clarity in the obtained results indicates the need to continue research on the importance of Modeling as a feeding practice intentionally used by mothers toward their children while considering the mother’s vegetable consumption as unintentional Modeling.

On the other hand, the results indicate that Modeling was next to mothers’ Monitoring and Involvement in the most used parental feeding practices, which might suggest their positive impact on fruit and vegetable consumption [21,40,41,42,43,44]. Although two-way positive correlations between these parental feeding practices, children’s fruit and vegetable intake, and the frequency of eating them were observed in the study group, there was no predictive effect of Modeling in explaining the recommended intake of fruit and vegetables. As mentioned earlier, this may be due to mothers’ relatively low consumption of vegetables and fruit, as only 27.3% consumed the recommended 4 portions of fruit and vegetables. In contrast, children’s fruit and vegetable consumption correlated negatively with using Food as a Reward, as also shown in other studies [45,46], which results primarily from an increased preference for the food used as a reward and a decreased preference for the food that was initially promoted [47,48]. At the same time, the negative relationship between Food as a Reward and fruit and vegetable intake was confirmed by the lower likelihood of meeting the recommended fruit and vegetable intake among children with parents having a higher use of this practice. It is known, however, that offering non-food rewards can increase children’s vegetable acceptance and intake [46].

Monitoring practices, as one of the most frequently used in the study group, increased the consumption of fruit and the likelihood of meeting the recommended fruit and vegetable intake in children, as confirmed by other studies [49,50]. However, covariates strongly influenced this effect. For fruit, maternal intake did not modify the predictive effect of monitoring children’s intake, whereas vegetable intake eliminated this effect. Moreover, in the group of older children, Monitoring increased the probability of consuming the recommended number of fruit and vegetable portions by about four times, regardless of including covariates in the analysis. In contrast, this effect was not observed in the group of younger children. In the group of younger children, a negative relationship of using Food as a Reward was found, while this practice did not explain the intake in children aged 6–10 years. The use of Food as a Reward may be because it has a variety of uses at a younger age, including enabling cooperation with the child in different situations, e.g., a visit to the doctor’s, during meals, or in the car [51]. Some previous studies demonstrated that children frequently rewarded with food tended to eat more snacks [52,53], more sugar-containing beverages, and less fruit [54], which aligns with our results.

### Strengths and Limitations

Although this is a cross-sectional study in which mothers report on their feeding practices and children’s eating habits, the results help us to better understand the links between parental feeding practices and fruit and vegetable consumption among children.

The study has certain limitations. The analysis relies on self-reported measurements, including fruit and vegetable intake, which should be considered biased. However, previous studies have shown consistent associations between parental reports and behavioral measures of eating among children [55,56]. Cross-sectional studies fail to track changes in fruit and vegetable consumption and capture the different challenges parents face as their children grow. Additionally, our study focused solely on a unidirectional relationship, positioning mothers as the primary influence. However, considering a bidirectional relationship might provide better insights into the complex interactions between mothers and children’s eating behaviors [57,58]. Another limitation of the study is that only maternal practices are considered, while many other family environment characteristics may be important, including breastfeeding the child. Previous studies have shown that breastfeeding can be an effective preventive intervention for children with a low intake of vegetables [59]. However, the questionnaire did not include a question on breastfeeding. Moreover, the role of other caregivers, e.g., fathers or other adult family members, and the impact of siblings on the child included in the study were not considered. The limitations of this study also include the recruitment to the study, the convenience sample, and the sample size. Lack of representativeness in terms of age, gender, education, and geographical location, etc., causes the outcomes not to be generalized to the whole population. In turn, convenient sampling may lead to bias towards dads with a higher fruit and vegetable intake. The small sample requires not only caution in interpreting the obtained results, but also a continuation of the study in a larger group that meets the requirements of representativeness for the population of Mazovia. Moreover, the data collection method used in the CAWI (Computer-Assisted Web Interview) survey is characterized by self-selection, which causes a potential source of bias. Those responding may differ significantly from other eligible individuals who chose not to answer the questionnaire. Despite these limitations, the results of this study may contribute to a better understanding of the relationship between mothers’ practices when feeding their children and children’s consumption of fruit and vegetables.

## 5. Conclusions

The results confirm that maternal feeding practices may influence children’s fruit and vegetable consumption. The Food as a Reward and Emotion Regulation practices were negatively associated with meeting fruit and vegetable recommendations in younger children, while the Monitoring practice was positively related to them being met in older children. These differences may indicate that maternal feeding practices do not equally and continuously influence children’s dietary behaviors. Furthermore, differences were separately observed in the associations between maternal feeding practices and fruit and vegetable consumption. The Monitoring practice correlated positively with fruit intake, while the Food as a Reward practice correlated negatively with vegetable intake in children. The mother’s fruit intake showed the strongest association with the child’s fruit intake, and a similar correlation was found for vegetable intake. The results can help develop and implement effective intervention strategies for a healthier diet among this age group.

However, the low consumption of fruit and vegetables among children, confirmed in the current study, prompts further investigation of the factors influencing children’s dietary behaviors (e.g., the importance of other family members and peers or marketing factors). Future research should also explore how the importance of diverse factors changes from 4 to 10 years. A longitudinal study with repeated measures is needed to replicate these findings and increase the understanding of children’s fruit and vegetable intake.

## Figures and Tables

**Table 1 nutrients-17-00941-t001:** Characteristics of the study sample (N = 260).

Socio-Demographic Characteristics	Total Sample % (N) *	*p*-Value **
Mother’s education		
Secondary	38.8 (101)	<0.001
Higher	61.2 (159)	
Place of residence		
A village	17.7 (46)	<0.001
A town with less than 100,000 inhabitants	22.3 (58)	
A city with over 100,000 inhabitants	60.0 (156)	
Mothers’ age		
30 years or less	18.8 (49)	
31–35	15.8 (41)	<0.001
36–40	34.6 (90)	
Above 40 years	30.8 (80)	
Child’s gender		
Girls	58.1 (151)	0.009
Boys	41.9 (109)	
Child’s age		
4–5 years	51.9 (135)	0.535
6–10 years	48.1 (125)	
Number of children in the family		
1 child	33.5 (87)	0.003
2 children	41.9 (109)	
3 children or more	24.6 (64)	

* N—number of participants; ** comparison of observed proportions and expected (equal) proportions for a variable—Chi-square goodness-of-fit test.

**Table 2 nutrients-17-00941-t002:** Consumption of fruit and vegetables in mothers and their children.

Frequency of Eating	Fruit					Vegetables				
	Mothers		Children		*p*-Value **	Mothers		Children		*p*-Value
	%	N *	%	N		%	N	%	N	
Less than once a month or never	4.2	11	3.8	10	0.214	3.1	8	3.5	9	0.207
1–3 times a month	1.5	4	6.2	16		3.1	8	6.5	17	
Once a week	14.2	37	9.2	24		17.6	46	22.7	59	
A few times a week	32.7	85	29.6	77		36.9	96	30.0	78	
Once a day	36.2	94	32.4	84		33.1	86	25.4	66	
A few times a day	11.2	29	18.8	49		6.2	16	11.9	31	

* Number of respondents; ** Wilcoxon signed-rank test.

**Table 3 nutrients-17-00941-t003:** Descriptive statistics of maternal feeding practices (MFPs) in the study sample.

Items	Maternal Feeding Practices (MFPs)					
	Emotion Regulation	Food as a Reward	Involvement	Modeling	Monitoring	Restrictions for Weight Control
Median	2.00	2.33	4.00	4.00	4.00	2.37
Minimum	1.00	1.00	1.00	1.00	1.75	1.00
Maximum	5.00	5.00	5.00	5.00	5.00	5.00

**Table 4 nutrients-17-00941-t004:** Correlations between maternal feeding practices (MFPs) used for their children and fruit and vegetable consumption among children.

	Intake		Frequency of Eating	
	Vegetables	Fruit	Vegetables	Fruit
**Parental Feeding Practices (PFPs) *****				
Emotion Regulation	−0.283 **	−0.090	−0.246 **	−0.076
Food as a Reward	−0.317 **	−0.167 *	−0.341 **	−0.155 *
Involvement	0.156 **	0.205 **	0.300 **	0.323 **
Modeling	0.278 **	0.286 **	0.465 **	0.396 **
Monitoring	0.287 **	0.396 **	0.465 **	0.500 **
Restrictions for Weight Control	0.065	−0.001	−0.077	−0.125 *

* Spearman’s correlation coefficient, *p* < 0.05; ** *p* < 0.01; *** a 5-point scale was used for each PFP.

**Table 5 nutrients-17-00941-t005:** Determinants of the consumption of vegetables in children—results of multiple linear regression models.

	Items	Non-Standardized Coefficients		Standardized Coefficients *		*p*-Value	95% CI
		B	Standard Error	β	t		
**Model 1 ****	Emotion Regulation ****	−7.842	8.295	−0.068	−0.945	0.345	−24.179; 8.494
	Food as a Reward	−22.815	6.865	−0.240	−3.324	0.001	−36.335; −9.296
	Involvement	−5.496	9.190	−0.042	−0.598	0.550	−23.596; 12.603
	Modeling	20.307	9.160	0.170	2.217	0.028	2.267; 38.347
	Monitoring	23.373	9.688	0.158	2.413	0.017	4.294; 42.453
	Restrictions for Weight Control	12.352	7.016	0.104	1.760	0.080	−1.466; 26.169
**Model 2 *****	Emotion Regulation	−13.203	7.660	−0.114	−1.724	0.086	−28.289; 1.883
	Food as a Reward	−15.623	6.392	−0.164	−2.444	0.015	−28.211; −3.034
	Involvement	−8.543	8.455	−0.065	−1.010	0.313	−25.194; 8.108
	Modeling	7.218	8.626	0.060	0.837	0.404	−9.771; 24.206
	Monitoring	4.511	9.310	0.030	0.485	0.628	−13.824; 22.846
	Restrictions for Weight Control	18.554	6.508	0.156	2.851	0.005	5.737; 31.371
	Mother’s vegetable intake	0.317	0.046	0.428	6.910	<0.001	0.227; 0.407

* Adjusted for mother’s age and gender of the child; ** Model 1—F = 10.015, *p* < 0.001, adjusted R^2^ = 0.173; *** Model 2—F = 16.992, *p* < 0.001, adjusted R^2^ = 0.302; **** a 5-point scale was used for each PFP.

**Table 6 nutrients-17-00941-t006:** Determinants of the consumption of fruit in children—results of multiple linear regression models.

	Items	Non-Standardized Coefficients		Standardized Coefficients *		*p*-Value	95% CI
		B	Standard Error	β	t		
**Model 1 ****	Emotion Regulation ****	9.237	8.061	0.083	1.146	0.253	−6.637; 25.112
	Food as a Reward	−10.227	6.671	−0.112	−1.533	0.127	−23.364; 2.910
	Involvement	3.668	8.931	0.029	0.411	0.682	−13.920; 21.256
	Modeling	12.747	8.901	0.110	1.432	0.153	−4.782; 30.277
	Monitoring	46.567	9.414	0.327	4.946	<0.001	28.027; 65.107
	Restrictions for Weight Control	−2.303	6.818	−0.020	−0.338	0.736	−15.730; 11.123
**Model 2 *****	Emotion Regulation	2.366	7.781	0.021	0.304	0.761	−12.958; 17.689
	Food as a Reward	−4.911	6.427	−0.054	−0.764	0.446	−17.569; 7.748
	Involvement	7.608	8.530	0.060	0.892	0.373	−9.192; 24.408
	Modeling	10.424	8.481	0.090	1.229	0.220	−6.279; 27.126
	Monitoring	31.452	9.410	0.221	3.342	<0.001	12.919; 49.985
	Restrictions for Weight Control	−3.422	6.490	−0.030	−0.527	0.598	−16.205; 9.360
	Mother’s fruit intake	0.305	0.058	0.309	5.241	<0.001	0.191; 0.420

* Adjusted for mother’s age and gender of the child; ** Model 1—F = 9.098, *p* < 0.001, adjusted R^2^ = 0.158; *** Model 2—F = 12,538, *p* < 0.001, adjusted R^2^ = 0.238; **** a 5-point scale was used for each PFP.

**Table 7 nutrients-17-00941-t007:** Fruit and vegetable intake in mothers and their children according to the number of consumed portions and the child’s age.

Groups	Intake of Fruit and Vegetables	Total Sample % (N) *	Children	
			4–5 Years Old % (N)	6–10 Years Old % (N)
**Total Sample**		100.0 (260)	100.0 (135)	100.0 (125)
**Mothers** (*p* < 0.001) **	4 portions or more	27.3 (71)	37.0 (50)	16.8 (21)
	Less than 4 portions	72.7 (189)	63.0 (85)	83.2 (104)
**Children** (*p* < 0.001) **	4 portions or more	49.2 (128)	64.4 (87)	36.0 (45)
	Less than 4 portions	50.8 (132)	35.6 (48)	64.0 (80)

* N—number of respondents; ** *p*-value (Chi-square test).

**Table 8 nutrients-17-00941-t008:** Odds ratios for recommended intake of fruit and vegetables in daily portions among children according to maternal feeding practices.

Maternal Feeding Practices	β	e^β^ (OR)	95 CI *		*p*-Value *
**The study sample—Model 1 ****					
Emotion Regulation ***	−0.311	0.732	0.449	1.195	0.212
Food as a Reward	−0.218	0.804	0.556	1.164	0.248
Involvement	0.402	1.495	0.926	2.413	0.100
Modeling	0.065	1.067	0.661	1.721	0.791
Monitoring	0.679	1.971	1.088	3.571	0.025
Restrictions for Weight Control	0.164	1.178	0.786	1.765	0.428
**Children aged 4–6—Model 2 ****					
Emotion Regulation	−1.064	0.345	0.157	0.760	0.008
Food as a Reward	−0.727	0.484	0.265	0.881	0.018
Involvement	0.190	1.209	0.605	2.418	0.590
Modeling	−0.310	0.734	0.351	1.533	0.410
Monitoring	0.263	1.300	0.447	3.781	0.630
Restrictions for Weight Control	0.586	1.796	0.877	3.678	0.109
**Children aged 7–10—Model 3 ****					
Emotion Regulation	0.168	1.183	0.567	2.466	0.655
Food as a Reward	−0.165	0.848	0.463	1.554	0.594
Involvement	0.332	1.394	0.562	3.460	0.474
Modeling	0.583	1.792	0.736	4.363	0.199
Monitoring	1.421	4.141	1.288	13.313	0.017
Restrictions for Weight Control	−0.336	0.715	0.384	1.330	0.289

* OR—point estimate (e^β^), 95% confidence intervals; significance level of Wald’s test, Models adjusted for mother’s age, child’s gender, and mother’s recommended intake of fruit and vegetables; ** Model 1—F = 12.920, *p* < 0.001, adjusted R^2^ = 0.317; Model 2—F = 7.679, *p* < 0.001, adjusted R^2^ = 0.310; Model 3—F = 3765, *p* < 0.001, adjusted R^2^ = 0.167; *** a 5-point scale was used for each PFP.

## Data Availability

The data presented in this study are available upon request from the corresponding author. The data are not publicly available because they have not yet been made available in ‘publicly available databases’.

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
