# Peer review of "Fruit and Vegetable Consumption in Mothers and Their Children Aged 4–10 and Its Relationship with Maternal Feeding Practices"

_nutrients, 2025, doi:10.3390/nu17060941_

Round 1

Reviewer 1 Report (New Reviewer)

Comments and Suggestions for Authors

Dear Authors:

Regarding the manuscript with title “Fruit and Vegetable Consumption in Mothers and their Children Aged 4-10 and its Relationship with Maternal Feeding Practices”, I have some minor comments to address.

Comment 1: Lines 50-52: “Moreover, the results of previous studies confirm the differences between mothers and fathers in parenting feeding practices and their impact on children's eating behavior [8-10].” Authors must specify the existing differences between gender.

Comment 2: Lines 50-52: “Moreover, the results of previous studies confirm the differences between mothers and fathers in parenting feeding practices and their impact on children's eating behavior [8-10].” This information must be transferred to lines 55-56 to be interconnected with the sentence “However, previous findings underline the importance of studying the individual role of each parent in child-feeding research [8] for a better flow of ideas.

Comment 3: Line 53: Authors have to contextualize the importance of the ingestion of fruits and vegetables before the reference to these two foods.

Comment 4: Why authors made this study with children aged 4-10? This question must be addressed on Introduction.

Comment 5: On Table 2, it is confusing the presentation of the p values. I suggest authors to add a column in the final of the Table with the p values. Also the intake in grams of vegetable plus fruit must be added to the Table as i tis referred in text by authors.

Author Response

Reviewer 1.

All changes in the manuscript are marked in yellow.

Regarding the manuscript with title “Fruit and Vegetable Consumption in Mothers and their Children Aged 4-10 and its Relationship with Maternal Feeding Practices”, I have some minor comments to address.

Comment 1: Lines 50-52: “Moreover, the results of previous studies confirm the differences between mothers and fathers in parenting feeding practices and their impact on children's eating behavior [8-10].” Authors must specify the existing differences between gender.

 Response to comment 1. Examples of such differences are given (i.e., fathers use more pressure to eat and fewer rewards, use more restrictions, and impose fewer restrictions on snacks).

Comment 2: Lines 50-52: “Moreover, the results of previous studies confirm the differences between mothers and fathers in parenting feeding practices and their impact on children's eating behavior [8-10].” This information must be transferred to lines 55-56 to be interconnected with the sentence “However, previous findings underline the importance of studying the individual role of each parent in child-feeding research [8] for a better flow of ideas.

 Response to comment 2. This information was transferred to be interconnected with the text.

Comment 3: Line 53: Authors have to contextualize the importance of the ingestion of fruits and vegetables before the reference to these two foods.

 Response to comment 3. As noted, this information has been moved to appear in the appropriate context. It seems after presenting the issue related to the consumption of fruits and vegetables and its determinants.

Comment 4: Why authors made this study with children aged 4-10? This question must be addressed on Introduction.

 Response to comment 4.In the Introduction we have included the sentence: ‘Since children's school activity and thus the long time spent outside the home and eating meals may influence the relationship between their fruit and vegetable consumption and parenting practices, being in kindergarten (age 4-6) and early school (age 7-10) was considered as a criterion for inclusion in the study’.

Comment 5: On Table 2, it is confusing the presentation of the p values. I suggest authors to add a column in the final of the Table with the p values. Also the intake in grams of vegetable plus fruit must be added to the Table as it is referred in text by authors.

Response to comment 5. The column with the p-values was added. Information on fruit and vegetable intake was removed from the table due to poor readability. The results were described in the text. Information on the intake of fruit and vegetables was also added.

We kindly thank the Reviewer for the time taken to review our article. We greatly appreciate all the comments and suggestions.

Reviewer 2 Report (New Reviewer)

Comments and Suggestions for Authors

To the Authors,

Thank you for the opportunity to review the original article entitled “Fruit and Vegetable Consumption in Mothers and their Children Aged 4-10 and its Relationship with Maternal Feeding Practices”.

Major comments:

Abstract—Conclusions: “The findings have shown the importance of mothers’ fruit and vegetable consumption in shaping their consumption in children.” These findings are too general and should focus on the main findings of the study.

Introduction – The introduction offers a background of the important role played by parents in shaping their children's eating habits. The authors presented the influence of early eating habits on future practices, especially related to fruit and vegetable preferences. The study aimed to “determine the relationship between the consumption of fruit and vegetables among children and their mothers and the maternal feeding practices mothers (MFP) used towards their children aged 4-10 years.” The authors presented the different roles played by parents in shaping eating habits, without providing more explanations of these differences. I consider that it would be important to describe with more detail the following affirmation: ”Moreover, the results of previous studies confirm the differences between mothers and fathers in parenting feeding practices and their impact on children's eating behavior [8-10].”

The Material and Methods section offers a clear description of the study design and the questionnaires applied to determine the food intake and practices related to children’s food intake control. However, more detailed explanations should be provided on the sample size determination and the reasons for selecting five schools and five kindergartens in Warsaw. Do the food practices in the capital or big cities differ from those in less vulnerable areas? Is the batch representative of the entire population? Were families from vulnerable groups also included? Did the sample include Warsaw or the whole Masovia province? In the results section, the authors stated that the participants came from “cities with over 100,00 inhabitants.”.

In the Results section, the authors described the socio-demographical characteristics of the participants, the frequency of eating fruit and vegetables among mothers and their children aged 4-10 years, the maternal feeding practices (MFP), and their relationship with fruit and vegetable consumption in the study sample, and the maternal feeding practices (MFP) and their relationship with meeting fruit and vegetable recommendations in children. In Table 2, the presentation of the intake of vegetables and fruit intakes in grams is not presented clearly. I also suggest presenting the p-value under the table as a legend. It would be interesting to introduce in the prediction model other factors related to fruit and vegetable intake, such as family income and level of education. This would provide a better understanding of parents' attitudes and behaviors related to their children's eating habits.

In Table 7, the data presented for mothers and their children's fruit intake is not intuitive. It would be clearer if they were separated visually or if a more structured format is used.

In the table, the Odds Ratio (OR) for monitoring is 1.971, indicating a 97.1% increase in the odds of adequate fruit and vegetable consumption for each unit increase in the monitoring score. However, the text mentions an increase of 8.8%, without being clear about the methodology by which this value was reached (line 241). The difference between the OR and this percentage raises questions about the interpretation of the results. It would be useful to clarify the calculation method used for this percentage, as ORs do not typically translate directly into absolute probabilities without a logit transformation or weighted average.

In the Discussion section, the authors used the abbreviation (V&F), which is not frequently used. Please use the full name instead of this abbreviation. The discussion section is well organized and offers interesting explanations of the results of the study.

In the limitation subsection, a discussion on the small sample size should be provided.

Author Response

Reviewer 2.

All changes in the manuscript are marked in green.

Thank you for the opportunity to review the original article entitled “Fruit and Vegetable Consumption in Mothers and their Children Aged 4-10 and its Relationship with Maternal Feeding Practices”.

Major comments:

Comment 1. Abstract—Conclusions: “The findings have shown the importance of mothers’ fruit and vegetable consumption in shaping their consumption in children.” These findings are too general and should focus on the main findings of the study.

Response to comment 1. We agree that these findings are too general and do not specifically address the purpose of the study. This has been changed. (The findings have shown that the mother’s fruit intake was the strongest predictor of children’s fruit intake. Moreover, some maternal feeding practices, i.e., Food as a reward and Emotion regulation practices, were negatively associated with meeting fruit and vegetable recommendations in younger children, while Monitoring practice was positively related to their meeting in older children).

Comment 2. Introduction – The introduction offers a background of the important role played by parents in shaping their children's eating habits. The authors presented the influence of early eating habits on future practices, especially related to fruit and vegetable preferences. The study aimed to “determine the relationship between the consumption of fruit and vegetables among children and their mothers and the maternal feeding practices mothers (MFP) used towards their children aged 4-10 years.” The authors presented the different roles played by parents in shaping eating habits, without providing more explanations of these differences. I consider that it would be important to describe with more detail the following affirmation: ”Moreover, the results of previous studies confirm the differences between mothers and fathers in parenting feeding practices and their impact on children's eating behavior [8-10].”

Response to comment 2. Examples of such differences are given (i.e., fathers use more pressure to eat and fewer rewards, use more restrictions, and impose fewer restrictions on snacks). However, we did not expand on this issue much because only mothers participated in our study.

Comment 3. The Material and Methods section offers a clear description of the study design and the questionnaires applied to determine the food intake and practices related to children’s food intake control. However, more detailed explanations should be provided on the sample size determination and the reasons for selecting five schools and five kindergartens in Warsaw. Do the food practices in the capital or big cities differ from those in less vulnerable areas? Is the batch representative of the entire population? Were families from vulnerable groups also included? Did the sample include Warsaw or the whole Masovia province? In the results section, the authors stated that the participants came from “cities with over 100,00 inhabitants.”.

Response to comment 3. The sample selection was convenient. On the one hand, easy access to respondents in Warsaw and its surroundings due to the authors’ place of residence, and on the other hand, the desire to ensure the homogeneity of the group in terms of place of living determined the choice of location. The selection of six schools and six kindergartens (five pairs in Warsaw) resulted from similar reasons. The assumption was that the school and kindergarten should be in one district. The presented results are part of a more extensive, ongoing study. The study's beginning was mainly conducted in Warsaw, so we accidentally omitted information on one pair of school-kindergarten from a smaller town near Warsaw. As for the following remark, it should be said that research on nutritional practices in large cities and other areas is limited. We were unable to obtain such results. Despite this, it can be assumed that such differences may appear, hence the decision to choose a relatively homogeneous area, i.e., Masovia region. The study group is not representative of the entire population of Masovia due to convenient selection. In principle, every mother whose child was in a given kindergarten and school could participate in the study because no additional inclusion or exclusion criteria were considered. The sample included Warsaw as an example of a large city. The statement that participants came from "cities with over 100,000 inhabitants" results from the fact that the questionnaire was also used in other studies, and such a category for large cities was used there.

Comment 4.The Results section, the authors described the socio-demographical characteristics of the participants, the frequency of eating fruit and vegetables among mothers and their children aged 4-10 years, the maternal feeding practices (MFP), and their relationship with fruit and vegetable consumption in the study sample, and the maternal feeding practices (MFP) and their relationship with meeting fruit and vegetable recommendations in children. In Table 2, the presentation of the intake of vegetables and fruit intakes in grams is not presented clearly. I also suggest presenting the p-value under the table as a legend. It would be interesting to introduce in the prediction model other factors related to fruit and vegetable intake, such as family income and level of education. This would provide a better understanding of parents' attitudes and behaviors related to their children's eating habits.

Response to comment 4. Table 2 was changed. The column with the p-values was added. Information on fruit and vegetable intake was removed from the table due to poor readability. The results were described in the text. Information on the intake of fruit and vegetables was also added. We agree with the comment that including other variables in the model would provide a better understanding of parents' attitudes and behaviors related to their children's eating habits. However, we do not have information on family income. In addition, the study group was not very diverse in terms of education, as it included only mothers with secondary and higher education; hence, such an analysis was not performed.

Comment 5. In Table 7, the data presented for mothers and their children's fruit intake is not intuitive. It would be clearer if they were separated visually or if a more structured format is used.

Response to comment 5. The table was slightly changed to be more precise. We hope the presentation is better now.

Comment 6. In the table, the Odds Ratio (OR) for monitoring is 1.971, indicating a 97.1% increase in the odds of adequate fruit and vegetable consumption for each unit increase in the monitoring score. However, the text mentions an increase of 8.8%, without being clear about the methodology by which this value was reached (line 241). The difference between the OR and this percentage raises questions about the interpretation of the results. It would be useful to clarify the calculation method used for this percentage, as ORs do not typically translate directly into absolute probabilities without a logit transformation or weighted average.

Response to comment 6. We apologize for this mistake. We don't know how it happened, and we didn't notice it when checking the text. It should be as the reviewer indicated. It was corrected.

Comment 7. In the Discussion section, the authors used the abbreviation (V&F), which is not frequently used. Please use the full name instead of this abbreviation. The discussion section is well organized and offers interesting explanations of the results of the study.

Response to comment 7. The abbreviation has been changed to fruit and vegetables. Thank you very much for your positive comment regarding the discussion.

Comment 8. In the limitation subsection, a discussion on the small sample size should be provided.

Response to comment 8. Such a discussion was provided in the limitation section.

We kindly thank the Reviewer for the time taken to review our article. We greatly appreciate all the comments and suggestions.

Reviewer 3 Report (Previous Reviewer 2)

Comments and Suggestions for Authors

This is a fascinating and well-written study describing how maternal eating practices and behavior influence their children’s fruit and vegetable intake.   The authors have addressed the reviewer's concerns, leading to a very solid manuscript.  I just have a few very minor suggestions at this point.

Line 60 is a bit awkward.  Perhaps, “Although improving….”

I think Table 5, 6 and 8 should have reminders in the Legend of how the Models differ

The authors should mention in the limitations that a convenience sample was used and may to lead a bias towards dyads with higher fruit and vegetable intake

Also, I would mention in the limitations that  the study may not be generalizable to other locations since it was done in one geographical location

In the conclusions, the abbreviation F&V is sometimes used without a previous definition.  I suggest using “fruit and vegetable” for consistency

Author Response

Reviewer 3

Changes in the manuscript are marked in grey.

This is a fascinating and well-written study describing how maternal eating practices and behavior influence their children’s fruit and vegetable intake.   The authors have addressed the reviewer's concerns, leading to a very solid manuscript.  I just have a few very minor suggestions at this point.

 Comment 1: Line 60 is a bit awkward.  Perhaps, “Although improving….”

Response to comment 1. Thank you for your positive comment. The suggestion was used to improve the text.

Comment 2: I think Table 5, 6 and 8 should have reminders in the Legend of how the Models differ.

Answer to comment 2. This information was added to the legend.

Comment 3: The authors should mention in the limitations that a convenience sample was used and may to lead a bias towards dyads with higher fruit and vegetable intake.

Answer to comment 3. This information was added to the limitation section.

Comment 4: Also, I would mention in the limitations that  the study may not be generalizable to other locations since it was done in one geographical location.

Answer to comment 4. This limitations was added to the limitation section.

Comment 5: In the conclusions, the abbreviation F&V is sometimes used without a previous definition.  I suggest using “fruit and vegetable” for consistency.

Answer to comment 5. The abbreviation has been changed to fruit and vegetables for consistency.

We kindly thank the Reviewer for the time taken to review our article. We greatly appreciate all the comments and suggestions.

Reviewer 4 Report (Previous Reviewer 3)

Comments and Suggestions for Authors

I have carefully reviewed “nutrients-3493608_ Fruit and Vegetable Consumption in Mothers and their Children Aged 4-10 and its Relationship with Maternal Feeding Practices”, which corresponds to the previous article “nutrients-3304825_ Family Factors Predicting the Implementation of Dietary Recommendations for Fruits and Vegetables among Polish Children Aged 4-10 Years”, to which modifications have been incorporated.

First of all, I appreciate the effort made by the authors and the clarifications provided. The new title aligns more closely with the study’s objective.

However, the article still has limitations concerning the sample size (260 women) and the recruitment method, which was conducted online without information control.

Abstract

The abstract initially refers to dietary interventions, whereas this study presents a cross-sectional snapshot that cannot establish causality. Therefore, it should instead describe the study as an analysis of the association between fruit consumption in mothers and their children aged 4 to 10 years.

In line 20, it is stated that maternal fruit intake predicts child fruit intake. This phrasing is inaccurate, as the study identifies an associated behaviour rather than a predictive relationship. The same issue appears in line 28.

Additionally, line 29 refers to children aged 6 to 10 years, whereas the study’s objective covers children aged 4 to 10 years.

The conclusion does not fully align with the study’s objective. The study does not demonstrate the importance of fruit and vegetable consumption, nor does it assess its effect. Therefore, I strongly recommend rewriting the abstract.

Introduction

In lines 57–58, the text discusses the impact of maternal dietary interventions on children’s intake. However, establishing such an effect would require a cohort study design. The introduction should instead emphasise the importance of understanding the association between maternal and child fruit and vegetable consumption to ensure that the study design appropriately aligns with its objective.

Materials and Methods

It is stated that the study sample was drawn from five kindergartens in Warsaw. It would be valuable to indicate the total number of children aged 4 to 10 years attending these institutions to understand what proportion of this population is represented by the study sample.

Regarding statistical analysis, when n > 50, the Kolmogorov-Smirnov test should be used to assess normality instead of the Shapiro-Wilk test.

In cross-sectional studies, the Odds Ratio (OR) is described as a measure of association, not causality or direct risk, since these designs do not establish the temporal sequence between exposure and outcome.

Results

  • In line 162, the median age should be reported along with a measure of variability, which in this case—given the non-parametric nature of the variable—should be the interquartile range (IQR).
  • Also in line 162, it is stated that most respondents lived in a city with approximately 100,000 inhabitants. This information would be more appropriate in the Materials and Methods section.
  • Table 1 should present a comparison of proportions for each category of the variables.
  • In Table 3, the mean and standard deviation are reported. Are these variables normally distributed?
  • In Table 4, correlation results are presented. To enhance interpretation, the scale used (PFP) should be clearly indicated. The same applies to Tables 5 and 8.

Discussion and Conclusion

The discussion should thoroughly address the study’s limitations.

The conclusion must align with the study’s objective and design. It should refer to associations or prevalence rather than prediction, as the latter implies causality, which this study does not establish.

Author Response

Reviewer 4.

All changes in the manuscript are marked in blue.

Comment 1: I have carefully reviewed “nutrients-3493608_ Fruit and Vegetable Consumption in Mothers and their Children Aged 4-10 and its Relationship with Maternal Feeding Practices”, which corresponds to the previous article “nutrients-3304825_ Family Factors Predicting the Implementation of Dietary Recommendations for Fruits and Vegetables among Polish Children Aged 4-10 Years”, to which modifications have been incorporated.

First of all, I appreciate the effort made by the authors and the clarifications provided. The new title aligns more closely with the study’s objective.

However, the article still has limitations concerning the sample size (260 women) and the recruitment method, which was conducted online without information control.

Response to comment 1. Thank you for your appreciation of our efforts to improve the manuscript. We took all the previous comments very seriously. We are aware of these study limitations. However, we hope that other elements of this study can compensate for these limitations, which we report in detail in the limitation section.

Abstract

Comment 2: The abstract initially refers to dietary interventions, whereas this study presents a cross-sectional snapshot that cannot establish causality. Therefore, it should instead describe the study as an analysis of the association between fruit consumption in mothers and their children aged 4 to 10 years.

Response to comment 2. This remark is a bit incomprehensible to us. Knowledge about all societal phenomena can be used in the actions taken. However, we agree that cross-sectional studies do not allow for the determination of causality. The relationship between variables can be used in further studies looking for causality. Still, it can also draw other people's attention to a possible explanation of the existing problem. We described the study as an analysis of the association between fruit consumption in mothers and their children aged 4 to 10 years.

Comment 3: In line 20, it is stated that maternal fruit intake predicts child fruit intake. This phrasing is inaccurate, as the study identifies an associated behaviour rather than a predictive relationship. The same issue appears in line 28.

Response to comment 3.This phrasing was changed in both places.

Comment 4: Additionally, line 29 refers to children aged 6 to 10 years, whereas the study’s objective covers children aged 4 to 10 years.

Response to comment 4. Separate analyses were conducted in groups of 4-6 years (kindergarten) and 7-10 years (early school). This information applies to the older group of children.

Comment 5: The conclusion does not fully align with the study’s objective. The study does not demonstrate the importance of fruit and vegetable consumption, nor does it assess its effect. Therefore, I strongly recommend rewriting the abstract.

Response to comment 5. Conclusions were rewritten. Now, they are focused on the objective of the study.

Introduction

Comment 6: In lines 57–58, the text discusses the impact of maternal dietary interventions on children’s intake. However, establishing such an effect would require a cohort study design. The introduction should instead emphasise the importance of understanding the association between maternal and child fruit and vegetable consumption to ensure that the study design appropriately aligns with its objective.

Response to comment 6. We believe it is possible to use such findings to explain what happens in the topic when the Introduction is written. We agree that we should focus on the relationships between variables. Still, the effect of the intervention (e.g., introducing or reinforcing a practice) in the form of changed behavior ultimately informs the relationship between variables.

Materials and Methods

Comment 7: It is stated that the study sample was drawn from five kindergartens in Warsaw. It would be valuable to indicate the total number of children aged 4 to 10 years attending these institutions to understand what proportion of this population is represented by the study sample.

Response to comment 7. Unfortunately, this information was not recorded at the time of the study, so it is difficult for us to estimate these indicators.

Comment 8: Regarding statistical analysis, when n > 50, the Kolmogorov-Smirnov test should be used to assess normality instead of the Shapiro-Wilk test.

Response to comment 8. We used the Kolmogorov-Smirnov test to assess the normality instead of the Shapiro-Wilk test.

Comment 9. In cross-sectional studies, the Odds Ratio (OR) is described as a measure of association, not causality or direct risk, since these designs do not establish the temporal sequence between exposure and outcome.

Response to comment 9. This comment was used to improve the manuscript. The word “prediction” was deleted from the manuscript.

Results

Comment 10. In line 162, the median age should be reported along with a measure of variability, which in this case—given the non-parametric nature of the variable—should be the interquartile range (IQR).

Response to comment 10. The information on the IQR was added.

Comment 11. Also in line 162, it is stated that most respondents lived in a city with approximately 100,000 inhabitants. This information would be more appropriate in the Materials and Methods section.

Response to comment 11. Since different solutions can be used to present the characteristics of the study sample, i.e. either in the Methods section or in the Results section, we decided to keep it in the Results section. We hope that it will be accepted.

Comment 12. Table 1 should present a comparison of proportions for each category of the variables.

Response to comment 12. In Table 1, the sample characteristics are presented. Unfortunately, we do not know how to compare the proportions for each category of the variables. 

Comment 13. In Table 3, the mean and standard deviation are reported. Are these variables normally distributed?

Response to comment 13. These variables were not normally distributed. Information on mean values was omitted. The changes in the manuscript were made.

Comment 14. In Table 4, correlation results are presented. To enhance interpretation, the scale used (PFP) should be clearly indicated. The same applies to Tables 5 and 8.

Response to comment 14. The information on scale was added.

Discussion and Conclusion

Comment 15. The discussion should thoroughly address the study’s limitations.

Response to comment 15. The study’s limitations were addressed in more detail.

Comment 16. The conclusion must align with the study’s objective and design. It should refer to associations or prevalence rather than prediction, as the latter implies causality, which this study does not establish.

Response to comment 16. Conclusions were rewritten to refer to associations. Now, they are focused on the objective of the study.

We kindly thank the Reviewer for the time taken to review our article. We greatly appreciate all the comments and suggestions.

Round 2

Reviewer 2 Report (New Reviewer)

Comments and Suggestions for Authors

To the Authors,

First, I want to congratulate you on your hard work in revising the original article, "Fruit and Vegetable Consumption in Mothers and their Children Aged 4-10 and its Relationship with Maternal Feeding Practices." The Introduction and the Material and methods Sections have been improved, providing a more sound structure for the article. The tables have been improved and are easier to read and understand.

I consider that the article has been substantially improved and could be considered for publication in Nutrients.

Author Response

To the Authors,

First, I want to congratulate you on your hard work in revising the original article, "Fruit and Vegetable Consumption in Mothers and their Children Aged 4-10 and its Relationship with Maternal Feeding Practices." The Introduction and the Material and methods Sections have been improved, providing a more sound structure for the article. The tables have been improved and are easier to read and understand.

I consider that the article has been substantially improved and could be considered for publication in Nutrients.

Response: Thank you very much for this nice response. 

Kind regards

Reviewer 4 Report (Previous Reviewer 3)

Comments and Suggestions for Authors

I have reviewed the latest version of the manuscript “nutrients-3493608_ Fruit and Vegetable Consumption in Mothers and their Children Aged 4-10 and its Relationship with Maternal Feeding Practices”as well as the authors' comments on the suggestions provided.

I appreciate the effort the authors have made to improve the quality of the article.

Regarding Table 1, as I previously mentioned, it would be useful to include a comparison of proportions between the categories using a chi-square test with a 95% confidence level. If you are unfamiliar with this analysis, you may wish to consult an epidemiologist or a statistician.

Author Response

Comment of the Reviewer

I have reviewed the latest version of the manuscript “nutrients-3493608_ Fruit and Vegetable Consumption in Mothers and their Children Aged 4-10 and its Relationship with Maternal Feeding Practices”as well as the authors' comments on the suggestions provided.

I appreciate the effort the authors have made to improve the quality of the article.

Regarding Table 1, as I previously mentioned, it would be useful to include a comparison of proportions between the categories using a chi-square test with a 95% confidence level. If you are unfamiliar with this analysis, you may wish to consult an epidemiologist or a statistician.

Response to comment

Dear Reviwer,

Thank you for this suggestion. We have included the proper changes in the table 1. 

Kind regards,

This manuscript is a resubmission of an earlier submission. The following is a list of the peer review reports and author responses from that submission.

Round 1

Reviewer 1 Report

Comments and Suggestions for Authors

This study investigated whether mothers experience in childhood and practice in parental feeding influenced their children’s dietary behavior of vegetables and fruits in Poland. Several subscales of experiences and practices were associated with intake of their children. They concluded experiences and practices influenced their children’s behavior. After adjustment for children’s sex and age in the models, however, they were not associated with their children’s intake. The children’s intake varied among their age. The results dd not support the conclusion. This conclusion should be revised.

Furthermore, mothers’ intake of fruits and vegetables were different depending on their children’s age (L237239). “There was also a higher percentage of mothers of younger children (37.0%) who had recommended fruit and vegetable intake compared to mothers of older children (16.8%) (Table 6).” The authors referred that children’s consumption decreased as age around seven years (L264268). However, there are no reason for the difference of intake of mothers with children’s age between <6, and 6 years. Bias of food intake could not make them interpret simple correlation analysis seen in Tables 4, and 5. Multivariate analysis adjusting for children’s age or stratified analysis with children’s age are required for conclusion.

L138 Explain whether “at least 4 portions (400g) of fruit and vegetables per day” is appropriate for the age of 4. Are fruits of 600 g and vegetables of 500 g healthy? Too much carbohydrate, and maybe deficiency other nutrients. There may be a reporting bias and there is no energy (total amount) adjustment. Outliers should be excluded, and the plausibility of analysis should be discussed.

L369 Lack of representativeness in terms of age, gender, education, etc. causes the outcomes not to be generalized to the whole population.” L61 “Vegetable and fruit consumption among children correlates with parent’s education level.” The authors obtained socioeconomical characteristics, mother’s education and residence place in Table 1. These variables should be adjusted in multivariate model in L149, and Table 7.

The introduction is highly matched with the previous report by the authors, as iThenticate indicates.

Minor points.

L84 “with at least one child aged 4-10” When the participants have 2 or more children, which children did the mother answer, or did the authors categorize? When participants had children less than 4, or older than 10 along with targeted children, how did the authors treat and interpret their influence?

L97 How many questions do the questionnaires (DVPQ, CFE, and Dietary Habits and Nutrition Beliefs Questionnaire for mother and child) have, and how long does it take to answer on Web?

L119 “This answer scored zero points and was further treated as a missing value.” How were the participants with a missing value treated? They were excluded from summing scores (it must be means), or imputation was used?

L131 Transformed frequency of food intake ( 02 times / days) should be replaced for an ordered category (16) in L130131. Transformed frequency of food intake is used in Table 3.

L149 Adjusted for children’ age and sex, but not mothers.

L174 Amount of food intake calculated as frequency multiplied by a portion size should be added in Table 3. Does the bottom half of Table 3 indicate them? If so, calculation and a portion size should be added in the footnote.

L212 “r=” is inserted before 0.396.

Table 6. Units are unknown.

Author Response

Reviewer 1. (marked in yellow)

Comment 1. This study investigated whether mother‘s experience in childhood and practice in parental feeding influenced their children’s dietary behavior of vegetables and fruits in Poland. Several subscales of experiences and practices were associated with intake of their children. They concluded experiences and practices influenced their children’s behavior. After adjustment for children’s sex and age in the models, however, they were not associated with their children’s intake. The children’s intake varied among their age. The results dd not support the conclusion. This conclusion should be revised.

Response to comment 1.

As suggested, the conclusions have been carefully revised. The type of study does not allow for the determination of impacts. Still, it provides for the relationship assessment between variables, which has been clarified in the revised version. The comments we received from all reviewers encouraged us to rethink our approach to the statistical analysis. Since our study aimed to estimate the association between parental feeding practices and a child’s recommended fruit and vegetable intake, we decided to include the mother's fruit and vegetable intake as a variable in the adjusted model rather than in the crude model. This approach yielded results that reflect the relationship between parental practices and the dependent variable, even after covariates were included (adjusted model). According to the results, the description of results, their discussion, and, above all, as suggested, the conclusions have been changed

Comment 2. Furthermore, mothers’ intake of fruits and vegetables were different depending on their children’s age (L237–239). “There was also a higher percentage of mothers of younger children (37.0%) who had recommended fruit and vegetable intake compared to mothers of older children (16.8%) (Table 6).” The authors referred that children’s consumption decreased as age around seven years (L264–268). However, there are no reason for the difference of intake of mothers with children’s age between <6, and ≥6 years. Bias of food intake could not make them interpret simple correlation analysis seen in Tables 4, and 5. Multivariate analysis adjusting for children’s age or stratified analysis with children’s age are required for conclusion.

Response to comment 2.

We agree that there are no reasons for the difference in intake of mothers with children between <6 and ≥6 years. Nevertheless, these differences can be explained, among other things, by the greater involvement of mothers in feeding younger children, both with a view to the expected outcome of good nutrition and to teach children correct eating behavior. Less pressure to achieve this when children become older may be due to the increased time older children spend at school or the mother's greater involvement in work activities, which may weaken the mother's willingness to use modeling, thereby increasing fruit and vegetable consumption. These factors were not included in the study, with the exception of modelling. Performing additional analysis confirmed the greater importance of modeling as a parenting practice for younger children

Modeling

Child’s age

Mean

N

Standard deviation

<6

3,8093

135

1,02724

6-10

3,5900

125

,88126

Total

3,7038

260

,96427

p=0.016 - the U Mann-Whitney test

As suggested, we attempted to use MANCOVA, which would have allowed for the characterization of differences in group means regarding a linear combination of multiple dependent variables while simultaneously controlling for the child's age as a covariate. However, we abandoned this analysis because the required assumptions were not met. After checking for Homogeneity of Variance-Covariance Matrices (Box's M Test <0.001), we decided to give up this analysis. We divided the study group into two groups, i.e. children aged 4-5 years and 6-10 years, and performed the analyses in the total sample and separately in both groups (lines 278-292).

Comment 3. L138 Explain whether “at least 4 portions (400g) of fruit and vegetables per day” is appropriate for the age of 4. Are fruits of 600 g and vegetables of 500 g healthy? Too much carbohydrate, and maybe deficiency other nutrients. There may be a reporting bias and there is no energy (total amount) adjustment. Outliers should be excluded, and the plausibility of analysis should be discussed.

Response to comment 3

According to the Polish recommendations, at least 5 portions (400g) of fruits and vegetables should be consumed among both preschoolers as well as in older children (https://isap.sejm.gov.pl/isap.nsf/download.xsp/WDU20160001154/O/D20161154.pdf; https://ncez.pzh.gov.pl/wp-content/uploads/2022/02/E-book-Zywienie-w-przedszkolach-w-praktyce.pdf). The policies promoting higher intake of fruits and vegetables from an earlier age may be due to increased childhood obesity in Poland. However, we agree that 600g of fruit intake can be excessive and unhealthy, primarily since the correct ratio between fruits and vegetable intake should be maintained, i.e., greater consumption of vegetables than fruits. We agree that including total energy intake would have been valuable for the analysis; however, in the current study, we have focused only on fruit and vegetable intake to reduce the number of questions in the survey. The self-report bias about fruit and vegetable intake was added as a study limitation (Lines 366-367). 

Comment 4. L369 “Lack of representativeness in terms of age, gender, education, etc. causes the outcomes not to be generalized to the whole population.” L61 “Vegetable and fruit consumption among children correlates with parent’s education level.” The authors obtained socioeconomical characteristics, mother’s education and residence place in Table 1. These variables should be adjusted in multivariate model in L149, and Table 7.

Response to comment 4.

Mother’s and child’s age, child’s gender, number of children, mother’s education, place of residence, and mother’s recommended intake of fruit and vegetables were used as covariates in models presented in Table 7 and Table 8 (except for child’s age) (Lines 157-159; 275-277; 287-288).

Comment 5. The introduction is highly matched with the previous report by the authors, as iThenticate indicates.

Response to comment 5.

A previously conducted study considered the dietary experiences of adults, which is common to both texts and, hence, similarities in description. In contrast, the objectives of the research and other variables differ.

Minor points.

Comment 6. L84 “with at least one child aged 4-10” When the participants have 2 or more children, which children did the mother answer, or did the authors categorize? When participants had children less than 4, or older than 10 along with targeted children, how did the authors treat and interpret their influence?

Response to comment 6

When a woman had two or more children aged 4–10, she was asked to select one child and provide answers regarding gender, age, parenting practices, and fruit and vegetable consumption for that child (Lines 89-92). We did not consider the other children's impact on the child the mother identified as a participant in the study.

Comment 7. L97 How many questions do the questionnaires (DVPQ, CFE, and Dietary Habits and Nutrition Beliefs Questionnaire for mother and child) have, and how long does it take to answer on Web?

 Response to comment 7.

The questionnaire contained 49 statements on parental feeding practices, 39 statements on mothers' childhood food experiences, and 16 questions on frequency and number of portions (8 questions on children and 8 questions on mothers). The survey lasted no longer than 15 minutes.

Comment 8. L119 “This answer scored zero points and was further treated as a missing value.” How were the participants with a missing value treated? They were excluded from summing scores (it must be means), or imputation was used?

Response to comment 8.

The participants with a missing value were excluded from summing scores (Line 127). In Table 5, the number of mothers reporting on their food experiences from childhood referring to particular feeding practices is given in brackets (Lines 245-246).

Comment 9. L131 Transformed frequency of food intake ( 0–2 times / days) should be replaced for an ordered category (1–6) in L130–131. Transformed frequency of food intake is used in Table 3.

Response to comment 9.

The change was made (Lines 138-141; 192-194).

Comment 10. L149 Adjusted for children’ age and sex, but not mothers.

Response to comment 10.

This information was added (Lines 157-159).

Comment 11. L174 Amount of food intake calculated as frequency multiplied by a portion size should be added in Table 3. Does the bottom half of Table 3 indicate them? If so, calculation and a portion size should be added in the footnote.

Response to comment 11.

This information was added in the footnote (Lines 192-194).

Comment 12. L212 “r=” is inserted before 0.396.

Response to comment 12.

It has been done (Line 227).

Comment 13. Table 6. Units are unknown.

Response to comment 13.

Units have been indicated in Table 6 (Line 257)

We kindly thank the Reviewer for the time and effort taken to read and review our article. We greatly appreciate all the comments and suggestions.

Reviewer 2 Report

Comments and Suggestions for Authors

This interesting study adds to the body of literature on eating practices. I have a few suggestions that I think would add to the manuscript.

I have some questions about the survey:

With the invitation,  was the invitation to biological mothers only, or did it include adoptive or other guardians defined as mother? What was done with families with multiple children?

Did you include maternal age in the survey?  That might be nice to include in the analysis.

I would consider making this section as a table or bullets:

Items from the questionnaire are 111 distributed into 5 subscales (Restrictions, Healthy Eating Guidance, Pressure and Food 112 Reward, Monitoring, and Child Control). Respondents were asked to report how fre- 113 quently different situations took place in their childhood (when they were 7-8 years old), 114 using a 6-point scale: 1– “never”; 2 – “rarely”; 3 – “sometimes”; 4 – “mostly”; 5 – “always”, 115 6 – „I don’t remember”. Moreover, they agreed with the sentences describing family habits 116 from the period of their childhood using a 6-point scale: 1 – “disagree”; 2 – “slightly disa- 117 gree”; 3 – “neither agree nor disagree”; 4 – “slightly agree”; 5 – agree”; 6 – „I don’t remem- 118 ber”. The answer “I don’t remember” was added to minimize the risk of recall errors. This 119 answer scored zero points and was further treated as a missing value [30]

With the tables, I think it would be easier to read if the first column aligned left rather than the center.

When testing for differences between mother and child, I would include a column for p value

With table 4, I suggest including that this is by Spearman Correlation testing

In the discussion I suggest that a discussion of epigenetics may play a role and maternal intake specifically during pregnancy might be a factor

In the limitations a statement about how the population sample may not be generalizable.   There is certainly self-selection bias at play with a social media invitation

Author Response

Reviewer 2. (marked in blue)

This interesting study adds to the body of literature on eating practices. I have a few suggestions that I think would add to the manuscript. I have some questions about the survey:

Comment 1. With the invitation,  was the invitation to biological mothers only, or did it include adoptive or other guardians defined as mother?

Response to comment 1.

In the invitation, we addressed women who have a child aged 4-10. Such inclusion criteria as being a biological or adoptive mother were not considered. It also did not consider whether the woman was in a formal or informal relationship or whether the partner/husband was the biological father of a child aged 4-10.

Comment 2. What was done with families with multiple children?

Response to comment 2.

Such families were included in the study. Table 1 includes information on the number of children in the family. If a woman had two or more children aged 4–10, she was asked to select one child and provide answers regarding gender, age, parenting practices, and fruit and vegetable consumption for that child (Lines 90-93).

Comment 3. Did you include maternal age in the survey?  That might be nice to include in the analysis.

Response to comment 3.

The age of the mother was included in the questionnaire. In addition to information on the medium age of women participating in the study, more detailed information on the number of women in the age ranges was added to Table 1 (Lines 173-177). The age of the women was used in the adjusted model as one of covariates (Tables 7 and 8).

Comment 4. I would consider making this section as a table or bullets:

Items from the questionnaire are  distributed into 5 subscales (Restrictions, Healthy Eating Guidance, Pressure and Food Reward, Monitoring, and Child Control). Respondents were asked to report how frequently different situations took place in their childhood (when they were 7-8 years old), using a 6-point scale: 1– “never”; 2 – “rarely”; 3 – “sometimes”; 4 – “mostly”; 5 – “always”, 6 – „I don’t remember”. Moreover, they agreed with the sentences describing family habits from the period of their childhood using a 6-point scale: 1 – “disagree”; 2 – “slightly disagree”; 3 – “neither agree nor disagree”; 4 – “slightly agree”; 5 – agree”; 6 – „I don’t remember”. The answer “I don’t remember” was added to minimize the risk of recall errors. This answer scored zero points and was further treated as a missing value [30]

Response to comment 4. 

A table presenting the subscales with their components and information about the responses was attached to the manuscript. However, we propose to include the Table as a Supplementary Table.

Comment 5. With the tables, I think it would be easier to read if the first column aligned left rather than the center.

Response to comment 5.

It has been changed according to this suggestion.

Comment 6. When testing for differences between mother and child, I would include a column for p value

 Response to comment 6.

We added such information in the table and footnote.

Comment 7. With table 4, I suggest including that this is by Spearman Correlation testing

Response to comment 7.

 The information on Spearman’s rank correlation coefficient was added.

Comment 8. In the discussion I suggest that a discussion of epigenetics may play a role and maternal intake specifically during pregnancy might be a factor

Response to comment 8.

This suggestion was considered in the Discussion section as the limitation of the study (Lines 381-383).

Comment 9. In the limitations a statement about how the population sample may not be generalizable.   There is certainly self-selection bias at play with a social media invitation.

Response to comment 9.

Such information about self-selection bias was added to the limitations (Lines 385-386).

We kindly thank the Reviewer for the time and effort taken to read and review our article. We greatly appreciate all the comments and suggestions.

Reviewer 3 Report

Comments and Suggestions for Authors

The article titled “nutrients-3304825_ Family factors predicting the implementation of dietary recommendations for fruits and vegetables among Polish children aged 4-10 years “ is submitted to the “Pediatric Nutrition “ section of the Special Issue “Eating Behaviors in Children and Teens“.

Regarding the title, it implies a causal relationship, while the study is based on a cross-sectional design. This can be somewhat misleading; as cross-sectional designs can only establish associations rather than causation. I would suggest rephrasing the title to better align with the design used.

The aim of this study was to assess the link between mothers’ childhood food experiences (CFE) and their use of parental feeding practices with their own children aged 4–10 years, as well as to examine the relationship between these practices and children’s fruit and vegetable intake. The objectives were twofold: to determine the association between mothers' childhood food experiences (CFE) and the parental feeding practices (PFP) they employ with their children aged 4–10 years, and to examine the relationship between children’s fruit and vegetable consumption and parental feeding practices (PFP).

In the abstract, the objective is to explore the relationship between parental behaviors and children's behaviors through a cross-sectional design. However, this objective would be more appropriate for a longitudinal design, so it may need revision. Additionally, the criteria for inclusion and exclusion of participating women should be explained to validate the relevance of the information provided. It would also be beneficial to clarify the specific data collected. The use of computer-assisted interviewing methods should be discussed to address potential information biases.

With regard to keywords, it would be advisable to check their alignment with Medical Subject Headings (MeSH) terms.

Materials and Methods: The authors employed a Computer-Assisted Web Interview (CAWI) technique on social media, which introduces a selection bias, as only voluntary participants with social media access were included, excluding those without social media accounts. Although a cross-sectional design was used, the objectives of the study are more aligned with a longitudinal approach. Respondents were asked to report on various family situations and feeding practices when feeding their children; however, this retrospective information may introduce recall bias.

The criteria used to establish children’s sociodemographic characteristics should be clarified, especially in cases where participants had more than one child within the specified age range. It is unclear how the index child was selected for the study.

Statistical analysis suggests that a nested case-control analysis was performed within a cross-sectional design. The adequacy of the sample size for the study’s objectives should be justified.

Results: In Table 1, further explanation is needed on how age and gender were determined in cases where multiple children fell within the study’s age range. In Table 2, comparisons between mothers' and children’s fruit and vegetable intake should be conducted, and an appropriate test should be applied.

The discussion section should begin with a general assessment of the results and connect these findings to relevant literature. Any unrelated literature should be removed, as it resembles a secondary introduction to the topic.

The authors recognize in the limitations section that a cross-sectional design is not ideal for addressing the study’s objectives.

The conclusion should focus on the study's contributions rather than summarizing the findings.

I find the topic highly engaging, as indicated by the extensive literature available. However, the methodological approach may not be the most suitable for these research aims.

Author Response

Reviewer 3 (marked in green).

The article titled “nutrients-3304825_ Family factors predicting the implementation of dietary recommendations for fruits and vegetables among Polish children aged 4-10 years “ is submitted to the “Pediatric Nutrition “ section of the Special Issue “Eating Behaviors in Children and Teens“.

Comment 1. Regarding the title, it implies a causal relationship, while the study is based on a cross-sectional design. This can be somewhat misleading; as cross-sectional designs can only establish associations rather than causation. I would suggest rephrasing the title to better align with the design used.

Response to comment 1.

We used the word ‘predicting’ mainly because logistic regression was used to assess the association between variables. Nevertheless, we agree with the comment that this may suggest causality. As suggested, the title has been modified to indicate an association. In addition, we have added information on the type of study (cross-sectional). The proposed title is “Family Factors linked to Implementing Dietary Recommendations for Fruits and Vegetables Among Polish Children Aged 4–10 Years – a cross-sectional study”.

Comment 2. The aim of this study was to assess the link between mothers’ childhood food experiences (CFE) and their use of parental feeding practices with their own children aged 4–10 years, as well as to examine the relationship between these practices and children’s fruit and vegetable intake. The objectives were twofold: to determine the association between mothers' childhood food experiences (CFE) and the parental feeding practices (PFP) they employ with their children aged 4–10 years, and to examine the relationship between children’s fruit and vegetable consumption and parental feeding practices (PFP).

In the abstract, the objective is to explore the relationship between parental behaviors and children's behaviors through a cross-sectional design. However, this objective would be more appropriate for a longitudinal design, so it may need revision. Additionally, the criteria for inclusion and exclusion of participating women should be explained to validate the relevance of the information provided. It would also be beneficial to clarify the specific data collected. The use of computer-assisted interviewing methods should be discussed to address potential information biases.

Response to comment 2.

Suggested changes have been made to the text. The formulation of the objective considers the relationships being explored between the variables, thus not implying causality (Lines 75-78). In addition, a clarification has been added regarding the recruitment of participants (Lines 87-91). The data collection method used the CAWI (Computer-Assisted Web Interview) survey, which is characterized by self-selection. It is a potential information bias, as reported in the limitations of the study (Lines 388-391).

Comment 3. With regard to keywords, it would be advisable to check their alignment with Medical Subject Headings (MeSH) terms.

Response to comment 3.

The alignment of keywords with Medical Subject Headings (MeSH) terms has been checked, and changes have been made.

Comment 4. Materials and Methods: The authors employed a Computer-Assisted Web Interview (CAWI) technique on social media, which introduces a selection bias, as only voluntary participants with social media access were included, excluding those without social media accounts. Although a cross-sectional design was used, the objectives of the study are more aligned with a longitudinal approach. Respondents were asked to report on various family situations and feeding practices when feeding their children; however, this retrospective information may introduce recall bias.

Response to comment 4.

These limitations are included in the text in the discussion section (Lines 369-371).

Comment 5. The criteria used to establish children’s sociodemographic characteristics should be clarified, especially in cases where participants had more than one child within the specified age range. It is unclear how the index child was selected for the study.

Response to comment 5.

A clarification has been added regarding the recruitment of participants (Lines 86-92). A woman who met the inclusion criteria, i.e. was over 18 years of age, was the mother of at least 1 child aged 4-10 years, and agreed to participate in the study was asked to identify the child (if there were two or more children in the family in this age range) whose fruit and vegetable intake, but also whose parenting practices were measured.

Comment 6. Statistical analysis suggests that a nested case-control analysis was performed within a cross-sectional design. The adequacy of the sample size for the study’s objectives should be justified.

Response to comment 6.

The invitation was addressed to women living in the Masovia province. Due to the lack of accurate information on the number of children aged 4-10 years, the number of such children was estimated. The number of children aged 0-14 in 2018 is 880949 children (N) (https://warszawa.stat.gov.pl/dla-mediow/informacje-prasowe/o-najwiekszym-skarbie-mazowsza-czyli-o-dzieciach,205,1.html). For each year of children, the same number of children was assumed, which was calculated as N/ 15 (number of years). The population of children aged 4-10 was estimated to be approximately 420,000. Using the sampling calculator  (https://www.naukowiec.org/dobor.html), a sample size of 267 was obtained (confidence level - 95%, fraction size - 0.5, maximum error 6%). Unfortunately, our sample size is 6 smaller due to recruitment problems and the need to terminate the survey. 

Comment 7. Results: In Table 1, further explanation is needed on how age and gender were determined in cases where multiple children fell within the study’s age range. In Table 2, comparisons between mothers' and children’s fruit and vegetable intake should be conducted, and an appropriate test should be applied.

Response to comment 7.

We only have information on the number of children in the family; no questions were asked about the age and gender of the child's siblings included in the study. This information has been added to Table 1. The comparisons between mothers’ and children’s fruit and vegetable frequency were conducted using the Wilcoxon signed-rank test and marked in green in Table 2 and Table 3. Detailed information is presented in the Table below.

Table Differences in frequency and intake of fruit and vegetables in pairs: mother and her child (Wilcoxon signed-rank test)

Items

Fruit

Vegetable

Mother versus her child

Mother versus her child

Frequency of eating

Z statistic

-1.047

-1.335

Significance

0.295

0.182

Intake (in grams)

Z statistic

-0.598

-7.264

Significance

0.550

<0.001

Comment 8. The discussion section should begin with a general assessment of the results and connect these findings to relevant literature. Any unrelated literature should be removed, as it resembles a secondary introduction to the topic.

Response to comment 8.

The suggestion was used to improve the discussion. Some literature sources have been removed. On the other hand, new sources have been added, a change necessitated by the modification of the statistical analysis method, i.e. the inclusion of new covariates.

Comment 9. The authors recognize in the limitations section that a cross-sectional design is not ideal for addressing the study’s objectives.

Response to comment 9.

We are aware of the limitations of this study due to its cross-sectional nature, as it does not allow us to describe changes in parenting practices and fruit and vegetable consumption. However, we did not have the opportunity to conduct a longitudinal study. This limitation is essential in interpreting the results, but careful interpretation can help identify and address this issue.

Comment 10. The conclusion should focus on the study's contributions rather than summarizing the findings.

Response to comment 10.

We appreciate your comment, which we are considering as we improve the text. Until now, reviewers have often asked us to focus on the results, influencing our approach to conclusions.

Comment 11.

I find the topic highly engaging, as indicated by the extensive literature available. However, the methodological approach may not be the most suitable for these research aims.

Response to comment 11.

We understand the concerns about the methodological approach but hope that how the manuscript has been revised will make the text acceptable.

We kindly thank the Reviewer for the time and effort taken to read and review our article. We greatly appreciate all the comments and suggestions.

Round 2

Reviewer 1 Report

Comments and Suggestions for Authors

1. Response 2. We agree that there are no reasons for the difference in intake of mothers with children between <6 and ≥6 years. Nevertheless, these differences can be explained, among other things, by the greater involvement of mothers in feeding younger children, both with a view to the expected outcome of good nutrition and to teach children correct eating behavior. Less pressure to achieve this when children become older may be due to the increased time older children spend at school or the mother's greater involvement in work activities, which may weaken the mother's willingness to use modeling, thereby increasing fruit and vegetable consumption. These factors were not included in the study, with the exception of modelling. Mothers with older children have less satisfaction with recommendations in Table 6. The underlined sentence is incorrect, and mis-interpreted. The older group does not indicate the future of the younger group. The two groups are not linked. This is a limitation in relation to a sampling bias, which should be added in L388 (“not to be generalized”).

2. Response 7. “The questionnaire contained 49 statements on parental feeding practices, 39 statements on mothers' childhood food experiences, and 16 questions on frequency and number of portions (8 questions on children and 8 questions on mothers). The survey lasted no longer than 15 minutes.” This information should be added to the Methods.

3. L27, as well as L358, and 403. “Intergenerational transmission of selected parental feeding practice” was not determined in this study. “The mothers’ experiences were linked to using same eating practices toward their children (L349351),” in which the former is the memory of mothers, and the latter is the behavior of mothers. It is not intergenerational transmission. Ref. 13 showed intergenerational transmission of dietary behaviors using three generation interviews. However, this study used only mothers’ memory and attitude, but not dietary behaviors of both sides. The conclusion should be revised.

4. The conclusion may be exaggerated (“confirm the potential role” in L28, and further research in L407). “Food as a reward” was significantly associated with younger-children’s intake, but not with older-children’s in Table 8. “Monitoring” was significantly associated with older-children’s, but not with younger-children’s. Mothers’ experience and practice did not equally and continuously influence children’s dietary behavior during the ages of 4 to 10 years. The advice to mothers may be “Don’t worry about your children’s intake of vegetables and fruits.” The suggestion for future research may be to explore other factors influencing dietary behavior and the change of associations during the ages of 4 to 10 years, rather than “to determine the role of parental feeding practices and the pathways between feeding practices and children's eating behavior to maximize the impact of nutrition interventions in changing children's eating behavior (L408410).” Furthermore, is the deficiency more important than the satisfaction with the recommendation? The conclusion should be revised.

5. For example, “Family” in the title is “mothers’.” “predicting” is “influencing” or “associated with” in the cross-sectional design. “Implementation” is “satisfaction” or “following,” because they do not recognize a recommended amount and how much to eat in usual life.

Author Response

Reviewer 1

Comment 1 (response 2). “We agree that there are no reasons for the difference in intake of mothers with children between <6 and ≥6 years. Nevertheless, these differences can be explained, among other things, by the greater involvement of mothers in feeding younger children, both with a view to the expected outcome of good nutrition and to teach children correct eating behavior. Less pressure to achieve this when children become older may be due to the increased time older children spend at school or the mother's greater involvement in work activities, which may weaken the mother's willingness to use modeling, thereby increasing fruit and vegetable consumption. These factors were not included in the study, with the exception of modelling.” Mothers with older children have less satisfaction with recommendations in Table 6. The underlined sentence is incorrect, and mis-interpreted. The older group does not indicate the future of the younger group. The two groups are not linked. This is a limitation in relation to a sampling bias, which should be added in L388 (“not to be generalized”).

Response to comment 1.

Our study does not allow us to explain these differences by referring to the results of our research, due to the nature of the study. We can try to explain the differences using the available knowledge. The results indicate that in the group of older children, both fewer mothers (16.8%) and fewer children (36.0%) met the recommended number of fruit and vegetable portions compared to the group of younger children (37.0% and 64.4%, respectively). However, an error occurred in our explanation. Instead of increasing, it should be decreasing. This was our oversight.

The explanation should be: “Nevertheless, these differences can be explained, among other things, by the greater involvement of mothers in feeding younger children, both with a view to the expected outcome of good nutrition and to teach children the correct eating behavior. Less pressure to achieve this when children become older may be due to the increased time older children spend at school or the mother's greater involvement in work activities. This may weaken the mother's willingness to use modeling, thereby decreasing her fruit and vegetable consumption. These factors were not included in the study, except modeling.

Our explanation is only a guess regarding the observed differences.

We agree that the two groups are not linked, which results from the nature of the study. The limitations already state that the study is a cross-sectional study (Line 373-374).

Comment 2 (response 7). “The questionnaire contained 49 statements on parental feeding practices, 39 statements on mothers' childhood food experiences, and 16 questions on frequency and number of portions (8 questions on children and 8 questions on mothers). The survey lasted no longer than 15 minutes.”

This information should be added to the Methods.

Response to comment 2.

This information was added to the Methods (Line 92).

Comment 3. L27, as well as L358, and 403. “Intergenerational transmission of selected parental feeding practice” was not determined in this study. “The mothers’ experiences were linked to using same eating practices toward their children (L349–351),” in which the former is the memory of mothers, and the latter is the behavior of mothers. It is not intergenerational transmission. Ref. 13 showed intergenerational transmission of dietary behaviors using three generation interviews. However, this study used only mothers’ memory and attitude, but not dietary behaviors of both sides. The conclusion should be revised.

Response to comment 3.

The use of this term was an error. Thank you for your comment. Changes have been made to the text.

 Comment 4. The conclusion may be exaggerated (“confirm the potential role” in L28, and further research in L407). “Food as a reward” was significantly associated with younger-children’s intake, but not with older-children’s in Table 8. “Monitoring” was significantly associated with older-children’s, but not with younger-children’s. Mothers’ experience and practice did not equally and continuously influence children’s dietary behavior during the ages of 4 to 10 years. The advice to mothers may be “Don’t worry about your children’s intake of vegetables and fruits.” The suggestion for future research may be to explore other factors influencing dietary behavior and the change of associations during the ages of 4 to 10 years, rather than “to determine the role of parental feeding practices and the pathways between feeding practices and children's eating behavior to maximize the impact of nutrition interventions in changing children's eating behavior (L408–410).” Furthermore, is the deficiency more important than the satisfaction with the recommendation? The conclusion should be revised.

Response to comment 4.

The conclusions have been revised. For this purpose, we used the suggestions contained in the review. We hope that we were able to take them all into account.

 Comment 5. For example, “Family” in the title is “mothers’.” “predicting” is “influencing” or “associated with” in the cross-sectional design. “Implementation” is “satisfaction” or “following,” because they do not recognize a recommended amount and how much to eat in usual life.

Response to comment 5.

Since these comments refer to the title, they were interpreted as a suggestion to change the title, which is why such a change was made. The title is “Maternal factors associated with the consumption of fruit and vegetables in children aged 4-10 – a cross-sectional study”

We kindly thank the Reviewer for the time taken to read our responses and review our article. We greatly appreciate all the comments and suggestions.

Reviewer 3 Report

Comments and Suggestions for Authors

I have carefully reviewed the latest version of the manuscript ““nutrients-3304825_ Family factors predicting the implementation of dietary recommendations for fruits and vegetables among Polish children aged 4-10 years“, as well as the authors’ responses to suggestions for enhancing the clarity of their work, and I appreciate the effort that has been invested.

The article is now more coherent, aligning better with the study’s design and its findings. However, there remains some difficulty in understanding the unit of analysis—is it the mother-child pair (in which case, only one mother-child pair is chosen from each family unit), or is it the mother herself, observed with different children, thus involving repeated measures of maternal characteristics across her children?

Additionally, as the information is self-reported and voluntarily provided, requiring access to social networks and retrospective recall, memory bias raises questions about the target population actually being examined. Furthermore, the sample is a convenience sample and significantly smaller than the population sample. All these methodological aspects introduce biases that should be acknowledged in the study’s limitations.

The conclusion is overly complex and should be made simpler and more concise.

Author Response

Reviewer 3.

I have carefully reviewed the latest version of the manuscript ““nutrients-3304825_ Family factors predicting the implementation of dietary recommendations for fruits and vegetables among Polish children aged 4-10 years“, as well as the authors’ responses to suggestions for enhancing the clarity of their work, and I appreciate the effort that has been invested. The article is now more coherent, aligning better with the study’s design and its findings.

Comment 1. However, there remains some difficulty in understanding the unit of analysis—is it the mother-child pair (in which case, only one mother-child pair is chosen from each family unit), or is it the mother herself, observed with different children, thus involving repeated measures of maternal characteristics across her children?

Response to comment 1.

The unit of the analysis is the mother-child pair. When a woman (our respondent) had two or more children aged 4–10, she was asked to select one child and provide answers regarding his/her gender, age, parenting practices, and fruit and vegetable consumption.  Thus, all the information we use in the analyses refers to the mother-child pair embedded in the family, but other family members are not considered.

Comment 2. Additionally, as the information is self-reported and voluntarily provided, requiring access to social networks and retrospective recall, memory bias raises questions about the target population actually being examined.

Response to comment 2.

Indeed, the participants reported (self-reported) the information and provided it voluntarily, which is the specificity of survey research. The Ethics Committee gives consent to this type of research if participation is voluntary, and the researcher provides the possibility of resigning from participation in the study. An invitation to participate in the survey was sent via social networks, which impacted the study group. However, it can be assumed that such networks are common among mothers of children aged 4-10. This is a limitation of the study, although any other method of invitation to the survey apart from a random selection from a closed list has such a limitation. Retrospective learning about the past is an accepted research technique, although it is burdened with memory errors. Therefore, the target group was mothers of children aged 4-10 who were active in social networks and agreed to participate in the study after learning about its purpose.

Comment 3. Furthermore, the sample is a convenience sample and significantly smaller than the population sample. All these methodological aspects introduce biases that should be acknowledged in the study’s limitations.

Response to comment 3.

The limitations section includes information concerning a convenience sample and a group size (Lines 383-385).

Comment 4. The conclusion is overly complex and should be made simpler and more concise.

Response to comment 4.

In line with your feedback, the conclusions have been simplified and we hope they are more concise.

We kindly thank the Reviewer for the time taken to read our responses and review our article. We greatly appreciate all the comments and suggestions.

Round 3

Reviewer 3 Report

Comments and Suggestions for Authors

I have reviewed once again the article titled “nutrients-3304825_ Family factors predicting the implementation of dietary recommendations for fruits and vegetables among Polish children aged 4-10 years “, as well as the authors' responses to the issues raised. 

I wish to commend the authors for their efforts in addressing the feedback, improving the analysis, and clarifying aspects that were previously unclear in the manuscript. I believe they have achieved a significant improvement. 

Although the idea is very interesting, the study still has notable limitations. These include the use of a snowball sampling method via social media, which prevents control over who participates and restricts participation to individuals active on social media. Additionally, the reliance on self-reported information is another key issue. 

I have raised these concerns on two previous occasions.